# The Effects of 15 or 30 s SIT in Normobaric Hypoxia on Aerobic, Anaerobic Performance and Critical Power

**DOI:** 10.3390/ijerph18083976

**Published:** 2021-04-09

**Authors:** Hakan Karabiyik, Mustafa Can Eser, Ozkan Guler, Burak Caglar Yasli, Goktug Ertetik, Aysegul Sisman, Mitat Koz, Tomasz Gabrys, Karol Pilis, Raci Karayigit

**Affiliations:** 1Faculty of Sport Sciences, Ankara University, Gölbaşı, Ankara 06830, Turkey; karabiyik@ankara.edu.tr (H.K.); mceser@ankara.edu.tr (M.C.E.); oguler@ankara.edu.tr (O.G.); burakcaglar90@gmail.com (B.C.Y.); goktugertetik@gmail.com (G.E.); aysegulsisman@mu.edu.tr (A.S.); mitat.koz@ankara.edu.tr (M.K.); 2Department of Physical Education and Sports, Iğdır University, Iğdır 76410, Turkey; 3Faculty of Sport Sciences, Kastamonu University, Kastamonu 37000, Turkey; 4Faculty of Sport Sciences, Muğla Sıtkı Koçman University, Muğla 48000, Turkey; 5Sport Centrum Faculty of Pedagogy, University of West Bohemia, 301 00 Pilsen, Czech Republic; tomaszek1960@o2.pl; 6Faculty of Health Science, Jan Dlugosz University, 42-200 Czestochowa, Poland; k.pilis@ajd.czest.pl

**Keywords:** HIIT, team athletes, normobaric hypoxia, chronic training, sprint duration

## Abstract

Sprint interval training (SIT) is a concept that has been shown to enhance aerobic-anaerobic training adaptations and induce larger effects in hypoxia. The purpose of this study was to examine the effects of 4 weeks of SIT with 15 or 30 s in hypoxia on aerobic, anaerobic performance and critical power (CP). A total of 32 male team players were divided into four groups: SIT with 15 s at FiO_2_: 0.209 (15 N); FiO_2_: 0.135 (15 H); SIT with 30 s at FiO_2_: 0.209 (30 N); and FiO_2_: 0.135 (30 H). VO_2max_ did not significantly increase, however time-to-exhaustion (TTE) was found to be significantly longer in the post test compared to pre test (*p* = 0.001) with no difference between groups (*p* = 0.86). Mean power (MPw.kg) after repeated wingate tests was significantly higher compared to pre training in all groups (*p* = 0.001) with no difference between groups (*p* = 0.66). Similarly, CP was increased in all groups with 4 weeks of SIT (*p* = 0.001) with no difference between groups (*p* = 0.82). This study showed that 4 weeks of SIT with 15 and 30 s sprint bouts in normoxia or hypoxia did not increased VO_2max_ in trained athletes. However, anerobic performance and CP can be increased with 4 weeks of SIT both in normoxia or hypoxia with 15 or 30 s of sprint durations.

## 1. Introduction

Two decades ago, sprint interval training (SIT), a new variation of high intensity interval training (HIIT) with repetition of short (30 s) all-out sprints emerged [1]. This training method is characterized by repeated maximal training bouts of approximately 30 s with 2–4 min rest-intervals [2]. SIT has been shown to be as efficient as traditional endurance training to increase muscle oxidative capacity with a much lower training volume [3]. Repeated SIT bouts evokes most of the physiological and biochemical systems used in aerobic efforts [4]. SIT induces changes in glycolytic enzyme levels, muscle buffering, and ionic regulation, resulting in enhanced anaerobic performance [1,3,5].

Following the emergence of SIT, in recent years, a new concept of SIT came forward. This concept was originally based on classical SIT performed in hypoxia (SIH) [6]. Performing training in hypoxic conditions has been used for decades to improve athletic performance at sea-level [7]. Originally, to improve sea-level athletic performance, altitude training camps were organized at moderate altitude (1800–2500 m) for 2–4 weeks and 2–3 times a year [6]. However, nowadays, training in simulated hypoxic environments created by masks or hypoxic chambers is more popular than traditional altitude training [8]. Such interventions are associated with limited expenses and travel restrictions for athletes who can stay in their home environment and sustain their ordinary lifestyle, while training a few days in a week under hypoxic conditions [9]. Related studies showed that training in hypoxia could induce non hematological adaptations such as upregulation of mitochondrial biogenesis [10,11] oxidative and glycolytic enzymes [11,12,13], monocarboxylate transporters [13,14], and angiogenesis [11]. In short, most studies emphasize that both aerobic and anaerobic components could benefit from hypoxic training [15]. The main mechanisms by which hypoxic training has additional effects compared to traditional sea level training is hypoxia inducible factor-1α (HIF- 1α), which upregulates angiogenesis and glycolysis in response to low levels of tissue oxygenation [16].

To date, most chronical training studies involving hypoxia have just examined the potential of traditional aerobic training [17]. On the basis of the aforementioned evidence, it is reasonable to foresee that SIT in hypoxia might be another pertinent exercise stimulus to boost endurance exercise performance at sea level. However, studies on this subject are limited so far [6,12,18]. Yet, to date studies shown that SIH has additional effects on training adaptations such as phosphofructokinase (PFK) activity levels and anaerobic threshold compared to SIT in normoxia (SIN) [12] but no improvements were seen in other aerobic and anaerobic performance parameters [6,12,18]. However, in order to understand the effects of SIH in more detail, further studies are required. These studies should differ in terms of rest and work-interval duration, hypoxic exposure duration, or hypoxia level. More studies are needed to compare effects of SIH and SIN on critical power (CP), which draws the border between heavy and severe exercise and is the indicator of the highest sustainable work rate of oxidative metabolism [19].

The work-interval duration of SIT is another important issue currently being studied [20]. Individual SIT bouts are characterized by a peak power output (PP) in the initial 10 s followed by a sharp decrease during the remaining 20 s [21]. Although clearly these kinds of efforts represent a robust training stimulus, whether the mechanism responsible is the initial 10 s (generation of peak power), the whole 30 s (attempted maintenance of a high power output), or the duration of the recovery period is uncertain [21]. Several studies have shown an increase in VO_2max_ following various HIIT and SIT protocols [21,22] and some other studies reported that performing SIT with work-interval durations of as low as 10–15 s improved aerobic and anerobic performance parameters [20,21,23].

To the best of the authors’ knowledge, several previous studies have simultaneously evaluated multiple SIT protocols to clarify the effect of decreasing SIT work-interval duration on aerobic performance [20,21,23,24]. However, no studies compared different work-interval durations (15 s vs. 30 s) in both hypoxic/normoxic conditions of SIT simultaneously. Therefore, the aim of this study was to compare the effect of 4 week short and long duration of SIT under normoxic and hypoxic conditions on aerobic and anaerobic performance.

## 2. Materials and Methods

### 2.1. Participants

Forty male team sport players between the ages of 19–24 who had not had an injury that prevented them from performing physical activity, participated in this study. However, in some parts of the study, 8 participants dropped out: 3 participants declared that they lacked time, while 5 participants left without any reason. Therefore, 32 total participants (football, handball, and basketball players) completed the outlined procedures (Table 1). The study was conducted at the end of the season so that participants did not participate in any extra training program except for the current study design. The training volume of the participants was 6 to 7 h/wk before the beginning of the study. All participants were healthy, nonsmoking lowlanders. Participants that had traveled to altitudes higher than 2000 m within the last 3 months prior to the beginning of the study were not included. Before participating in the study, the test and training protocols, as well as possible risks that may occur during the test-training sessions, were explained to the participants in detail. Participants were informed to avoid consuming supplements, caffeine or alcohol at least 24 h before the tests and training sessions. Also, they were asked to avoid strenuous physical activities before test-training days. Further, participants were asked to record and replicate a 24-h dietary intake prior to each testing session before and after the 4 weeks training period. In the first interview with the participants, written informed consent outlining the purpose, procedures, and risks of the protocol was obtained from all participants, and the study procedures followed the principles outlined in the Declaration of Helsinki and were approved by Ankara University, Non-interventional Clinical Ethics Committee (11-615-17).

### 2.2. Study Design

Participants were counter-balanced and randomly divided into four groups according to their pre VO_2max_ values in a single blind manner: 15 s * 4–6 rps * 4 min (sprint time * repeats * recover between sprints) at 1015 m (15 N: *n* = 8; FiO_2_: 0.209) or 3536 m (15 H: *n* = 8; (FiO_2_: 0.135) altitude; 30 s * 4–6 rps * 4 min, at 1015 m (30 N: *n* = 8; FiO_2_: 0.209) or 3536 m (30 H: *n* = 8; FiO_2_: 0.135) altitude. Normoxia altitudes were determined based on the the altitude at which training and test sessions were performed in Ankara, Turkey (890 m). Test sessions conducted before and after the 4 weeks training period started with VO_2max_ test (day 1), followed by a repeated wingate test (day 2) and critical power test (day 3). At least 24–48 h of rest was given between each measurement day. Body composition (lean body mass and body fat percentage) was measured via Jawon Segmental Analyzer (Korea). After the pre-tests, the training sessions were carried out every other day for 4 weeks, 3 times a week, and a total of 12 training sessions were performed on cycle ergometer. In order to provide the specified altitude, the mask connected to the Everest Summit II—Altitude Generator (Hypoxico, USA) was worn on the participants and was kept on during all the training sessions. Total training session was 23–24 min in the first 2 weeks and 32–33 min in the third and fourth week. To verify that hypoxic and normoxic conditions were delivered, oxygen was assessed by a pulse oximeter (Hypoxico Oxycon, USA) attached to the participants’ fingers before and after each sprint bouts. Pre and post tests was performed in a laboratory a 890 m altitude. Following a 4 weeks training period, the same test procedures with same order and time of the day were replicated for each participants.

### 2.3. VO_2max_ Test

After determining the body composition in the laboratory on the same day, the Bruce Protocol was carried out on the treadmill, using a Jaeger Masterscreen CPX model ergospirometer system (Germany) gas analyzer to determine the maximal oxygen consumption (VO_2max_). Immediately after the end of the test, lactate measurements were analyzed with the Lactate Plus (Nova Biomedical, USA) instrument. Borg Scale (6–20) was used to determine the rate of perceived exertion (RPE) level at each increment stage of the VO_2max_ test. The mean (RPEmean) and the highest value (RPEpeak) of these values were recorded. The test period to find the VO_2max_ was divided into 30-s segments and the mean of the data were taken, while the data of the highest 30-s segment were recorded as VO_2max_, and the mean of the VO_2_ values obtained during the whole test were recorded as VO_2mean_ [3]. In addition, the heart rate (HR) taken (Polar Team 2 telemetric system, Finland) at the VO2peak point was recorded as HRmax.

### 2.4. Repeated Wingate Test

On the second test day, a repeated wingate anaerobic power test was performed on a Monark894E (Monark Exercise AB, Vansbro, Sweden) cycle ergometer. Seat and handle positions were adjusted for each participant and this was recorded and replicated in before and after 4 weeks of training sessions. Following a 5 min warm-up at 60 watt with 5 s sprints without resistance on the second and third minutes, participants performed 4 × 30-s Wingate “all-out” sprints with 4 min of passive rest [21]. The test was automatically initiated by Monark test software when participant reached ≥150 rpm during unloaded pedaling and subsequent instant application of load corresponding to 7.5% of body weight (6). They were asked to remain sitting and cycle maximally during 30-s sprints. Peak power (PP), mean power (MP), and work done (kJ) were calculated with Monark Anaerobic Test Software (Version 3.3.0.0, Vansbro, Sweden). To make them more clear, data were presented as average of four sprints. Sprint decrement (sDec) was also calculated via following formula; Fatigue = (100 × (total sprint power/ideal sprint power))—1; a-) Total sprint power = average of all sprint power b-) Ideal sprint power = sprint number x highest sprint power [25]. Before and immediately after each sprint, muscle pain perception (MPA) with a visual analog scale (VAS), HR, and RPE were measured and presented as an average of four sprints.

### 2.5. Critical Power Test

The Critical Power test on the 3rd measurement day was performed using the Monark 894E cycle ergometer, as previously described [19]. In the warm-up period, participants cycled for 5 min at 70 watts. After a warm-up session, participants rested for 5 min. During the rest, participants were reminded of the rules to follow during the test, to perform their best performance and to cycle at the highest speed he could. The test consisted of two 3-min rounds of cycling. In the first 3 min, the athlete cycled at 70 ± 5 rpm without load. In the last 5 s of the first 3 min without load, participants were asked to cycle up to 110 rpm and with the “start” command, the second 3 min began and they needed to cycle at their highest speed. At the beginning of the second 3 min, 0.03 kg/body weight was applied. During the test, participants were supported by verbal motivation. After the test, the Critical Power data was found by calculating the average of the last 30 s of the power output recorded in the Monark Anaerobic Test Software (Version 3.3.0.0).

### 2.6. Training Procedures

Participants were randomly divided into four groups in a single blind manner: 15 s * 4–6 rps * 4 min (sprint time * repeats * recover between sprints) at 1015 m (15 N) or 3536 m (15 H) altitude; training at 30 s * 4–6 rps * 4 min, 1015 m (30 N) or 3536 m (30 H) altitude. After the pre-tests, the training sessions were carried out every other day for 4 weeks, 3 times a week, and a total of 12 training sessions were performed. The first training session was performed at least 24 h after the pre-tests. All training sessions were performed against 7.5% of the body weight of the participants when reaching ≥150 rpm during unloaded pedaling in the Monark 894E cycle ergometer. In order to provide the specified altitude, the mask connected to the Hypoxico Everest Summit II—Altitude Generator was placed on the participants and was kept on during the training. The warm-up and cooling protocol before and after the training was applied at 60 W for 5 min in total, and 5-s no-load sprints were made in the middle of the 2nd and 3rd minutes of the warm-up period. At the beginning and end of the each sprint bout, ratings of perceived exertion (RPE), muscle pain status, and oxygen saturation in the tissues were measured with the BORG (6–20), VAS and the oximeter (Oxycon, USA) attached to the participant’s finger, respectively. Heart rate (HR), peak power (PP) (w/kg), mean power (MP) (w/kg), Work (kJ), and the percentage decrease (%) of each sprint bout were taken from the Monark Anaerobic Testing Software (Version 3.3.0.0) and recorded. In addition, sDec was calculated with the formula used in the repeated wingate test [25]. The data obtained in each training was recorded by averaging at the end of 12 training sessions (Figure 1).

### 2.7. Statistics

Data were analyzed using Statistical Package for the Social Sciences (SPSS) software version 22.0 for Windows (SPSS, Chicago, IL, USA). The mean comparisons of the values obtained at different times were evaluated with Repeated Measures ANOVA to determine whether there was any difference between the data obtained from each test within and between groups, and if there was a difference, the Bonferroni Correction was used to understand which group it originated from. One-way ANOVA was used to evaluate the training data. All analyses resulting in *p* < 0.05 were considered to be statistically significant. The effect size is given as partial eta squared (η2).

## 3. Results

### 3.1. VO_2max_ Test

After four weeks of SIT, VO_2mean_ (F_1,7_ = 1.17, *p* = 0.31, η^2^ = 0.14) and VO_2max_ (F_1,7_ = 1.01, *p* = 0.34, η^2^ = 0.13) values did not significantly increase and no group x time interaction was found for VO_2mean_ (F_3,21_ = 1.39, *p* = 0.27, η^2^ = 0.17) and VO_2max_ (F_3,21_ = 1.35, *p* = 0.29, η^2^ = 0.16). In all groups, TTE in the post test (F_1,7_ = 50.17, *p* = 0.001, η^2^ = 0.88) increased statistically compared to pre-test values. The increase in TTE suggests an improvement in aerobic capacity and no difference was found in group x time interaction (F_3,21_ = 0.40, *p* = 0.76, η^2^ = 0.05). All related values can be seen in Table 2, Figure 2 and Figure 3.

RPEmean (*p* < 0.05) values recorded during the post test decreased compared to the pre test in all groups. There was no difference in RPEpeak and HR during VO_2max_ test in all groups. This situation can be accepted as both a physiological and perceptual indicator that the participants in the pre and post test performed with their maximals. It is seen that the LA taken after the test reaches higher values in the post-tests compared to the pre test values (*p* < 0.05) in all groups. This suggests an improvement of lactic acid tolerance. Moreover, it becomes more meaningful when considered together with the exhaustion time (Table 2).

### 3.2. Repeated Wingate Test

In the repeated wingate test, a statistical difference was observed in all values except HR data and sprint decline data. PP (F_1,7_ = 75.00, *p* = 0.001, η^2^ = 0.92) and MP (F_1,7_ = 36.07, *p* = 0.001, η^2^ = 0.84) significantly increased with 4 weeks of SIT from pre to post test in all groups with no difference between PP (F_3,21_ = 0.30, *p* = 0.82, η^2^ = 0.04) and MP (F_3,21_ = 0.54, *p* = 0.66, η^2^ = 0.07). Increases in PP and MP caused an increase in the total work (F_1,7_ = 23.38, *p* = 0.002, η^2^ = 0.77) in all groups. The significant decrease seen in sDec (F_1,7_ = 34.74, *p* = 0.001, η^2^ = 0.83) data in all groups is an indicator of the faster recovery between repetitions and the other sprint was performed with a reduced performance decrease. In addition, there were significant (*p* < 0.05) decreases in the values of MPA and RPE at both the starting points and the end points of sprints. The decreases in the starting point also support the idea that the Sdec data is collected rapidly. The absence of a difference in HR average of post sprint (F_1,7_ = 1.05, *p* = 0.34, η^2^ = 0.13) values indicates that the test was performed by creating the same physiological responses.

### 3.3. Critical Power Test

CP was also sigificantly increased in the post test in all groups (F_1,7_ = 38.36, *p* = 0.001, η^2^ = 0.85) with no difference between groups (F_3,21_ = 0.31, *p* = 0.82, η^2^ = 0.04). All related parameters can be seen in Table 3 and Figure 4 and Figure 5.

### 3.4. Training

In the training data, there was no difference between the groups in PP (*p* = 0.24), but it was observed that 15 H and 15 N groups reached higher MP values than 30 H and 30 N groups (*p* = 0.001). Work in kj was found to significantly higher in 30 H and 30 N groups than in 15 H and 15 N groups (*p* = 0.001). It was observed that the percentages of decline in both sprint decrement (*p* = 0.001) and sDec data (*p* = 0.001) were sharper in the 30 H and 30 N groups than in the 15 H and 15 N groups. The similar adaptations in all parameters after 4 weeks of SIT with 15 s and 30 s of sprint bouts make the reasons for the length of the 30-s sprints questioned. There was no difference between the groups in MPA, RPE, and HR values (*p* > 0.05). Oxygen saturation was higher in the normoxic group compared to the hypoxic group (*p* = 0.001). This value was also of great importance in terms of seeing that the training creates the desired hypoxic effect. All of these results can be seen in Table 4.

## 4. Discussion

The main purpose of this study was to examine the effect of SIT with short (15 s) and long (30 s) work-interval durations on aerobic and anaerobic performance parameters in hypoxic and normoxic conditions. The originality of this study is that it is the first study to examine the duration of the work-interval of SIT in normobaric hypoxia. Our study has two major findings. One of these finding is that there was no difference between hypoxia and normoxia groups in aerobic and anaerobic training adaptations. Aerobic and anaerobic capacities in both SIN and SIH groups have been similarly improved. These data contradict our hypothesis that the additional physiological stress of SIH increases aerobic contribution during recovery, resulting in further training adaptations. Another finding of the current study is that decreasing the sprint duration from 30 s to 15 s did not negatively affect the aerobic and anaerobic training adaptations.

Theoretically, SIH may increase aerobic capacity, which is most often assessed by maximal oxygen uptake (VO_2max_), as well as improve sea-level endurance performance through varied training adaptations [26]. However, recent research findings about SIH as a more effective method than SIN for improving aerobic capacity and anaerobic performance variables at sea-level are insufficient. In a study with healthy males, after 6 weeks, SIH upregulated muscle PFK activity and the anaerobic threshold more than SIN [12] but there was no difference in VO_2max_, TTE, and blood lactate levels between groups. Richardson and Gibson [27] in their study with healthy males, after 2 weeks of training (30 s work/4 min rest) in moderate hypoxia (FiO_2_: 0.15), observed positive improvement but no difference between groups in VO_2peak_, TTE, and resting HR parameters. In a study [7] conducted on team sport players, researchers investigated the use of moderate to high intensity hypoxic cycle training in normobaric hypoxia (~3000 m) over 4 weeks. VO_2peak_ onset of blood lactate accumulation (OBLA), mean power, and peak power increased after both normoxic and hypoxic training, yet no differences were observed among normoxic and hypoxic training conditions. Moreover, 3 weeks of normobaric hypoxic endurance training did not significantly increase VO_2peak_ but did enhance test duration and distance [26]. In another study [28], without a difference between hypoxia and normoxia groups, an improvement of swimming performance and VO_2max_ in male and female swimmers was observed after 5 weeks of hypoxic training program in a flume, including short (30 s × 10 repetition) intervals with high intensity. In our study, VO_2max_ levels show no change after a 4 week (in total 12 sessions of SIT) training program. The fact that our participants have a training background and a high baseline VO_2max_ level (~60 mL.kg^−1^.min^−1^) may have caused this situation. Thus, 12 sessions of SIH and SIN may have not provided enough additional training stimuli to increase VO_2max_ in this population. Future studies should investigate the same training protocol at the beginning of the season when participants have lower VO_2max_ levels to observe the effect of training on this parameter. Longer duration studies reported significant increase in VO_2peak_ from ~51–54 mL.kg^−1^.min^−1^ after 7 weeks of SIT. Another study [29] demonstrated that VO_2peak_ improved following 6 weeks of SIT. It can be speculated that more than 4 weeks of SIH training may need to improve the VO_2peak_ values of participants that already have a high baseline level.

The aforementioned studies were conducted at approximately 3000 m. These results may differ at higher levels of hypoxia. Indeed, in a study [6] conducted at different altitudes (between FiO_2_ = 13.0–20.9) found them to provide higher PP and lactate-related adaptation gains than the same training routine at sea level, while training at lower altitudes did not provide additional gains. Authors of the research suggest that this increase in PP was probably not related to energy metabolism, but to muscle structure or the recruitment of fast-twitch fibers. Previous studies showed that higher altitude conditions may have induced a higher expression of fast-twitch fibers. Takei et al. [9] observed an increase in PP during repeated 30 s all-out efforts after two weeks (three sessions per week) of both SIH and SIN in similar conditions. Our findings are similar to those of this study. To our knowledge, in most of the studies performed at this altitude (2500–3000 m), no additional increase in performance parameters was observed during the wingate test after short- or long work-interval training protocols compared to normoxia [6,30].

In contrast to VO_2peak_, TTE increased in all groups. Previously, increases in TTE were associated with improvements in mitochondrial function after 2 weeks of wingate-based SIT or 8 weeks of aerobic HIIT at ~90% VO_2max_ [31]. Additionally, faster VO_2max_ kinetics were also related to the increase in TTE after 8 weeks of HIIT or 2 weeks of wingate-based SIT. Accelerated VO_2_ kinetics may delay the onset of depletion of muscle phospocreatine (PCr) and accumulation of fatigue-related metabolites (Pi, H+) [32,33], which likely results in an improved training tolerance [34]. In addition, Daussin et al. also reported an increased TTE in a traditional endurance training group (25–30 min cycling at ~60% VO_2max_) in the absence of mitochondrial adaptation but due to greater enhancements in vascular conductance and capillary density compared to the HIIT group [31]. This indicates that improved muscle perfusion and thus O_2_ supply in addition to improved muscle O_2_ uptake may increase the tolerable training period.

Another important finding of our study is that shortening the work-interval duration of SIT has no diminishing impact on aerobic and anaerobic training adaptations. Our choice of short sprint duration (15 s) was made based on the previous studies showing that during the maximal sprint, most of the anaerobic metabolism (i.e., the degradation of PCr and muscle glycogen) occurs within the first 15 s [20]. Studies that using SIT protocols similar to the ones in our study (repeated maximal 30 s efforts separated by 4 min recovery) have caused significant improvements in both anaerobic and aerobic power [35], Increases in glycolytic and oxidative enzymes activity [1], muscle buffering capacity [3], and ionic regulation [36] have been implicated in these results. As in these studies, significant increases were observed in all groups in anaerobic performance parameters such as absolute and relative PP and MP, as well as total work after training in our study. In addition, during the VO_2max_ test, similar blood lactate accumulation levels were recorded in each phase test for all groups. Additionally, RPE and MPA values which indicate participants’ perceptions about the strenuousness of tests were recorded. Results showed that after the training program, participants’ RPE and MPA values for each phase of VO_2max_ test were similar. These similar findings about RPE, MPA, blood lactate, and PP-MP support a similar anaerobic demand for both SIH and SIN sprint protocols. In addition to these results, our findings confirm those previously reported by Hazell et al. [21], Zelt. et al. [23], and Yamagishi et al. [20] that decreasing the work-interval duration of SIT by 50% during a four-week training intervention does not diminish adaptations of anaerobic and aerobic capacity. We have also examined the effects of the decreasing work interval SIT volume on training-induced increases in CP. In our study, it was observed that the CP data increased significantly (>15%) in all groups. This improvement in CP means that athletes can perform at higher intensity without getting tired during training or competition and this result is consistent with previous training interventions demonstrating significant increases in CP following varied HIT and SIT protocols [19,37]. Based on this information, we can clearly state that decreasing work-interval durations in SIT has no diminishing effect on aerobic and anaerobic performance adaptations. Therefore, it may be suggested that coaches and trainers should prefer short (15 s) work-interval duration SIT rather than long (30 s) work-interval duration SIT protocols in order to reduce total training volumes. Further, although perceived training enjoyment and psycho-physiological parameters such as pleasure, recovery, and breathlessness [38] were not measured during each SIT bouts in the current study, athletes can be motivated towards 15 s SIT protocols more easily than 30 s to increase aerobic and anaerobic performance.

Our study has two main limitations. One of the important limitations of the current study is a lack of a hypobaric environment. The findings of the study in a hypobaric environment could vary [39]. Changes in pressure according to the simulated altitude level, as in real altitude training, could affect training adaptations. Another limitation is that the nutrition of subjects is not controlled during the training program. Participants were warned not to use supplements that could affect the results 24 h before pre and post tests but their compliance was not checked. Nutrition is an important component of adaptations to training. Therefore, monitoring the nutrition of the participants could have strengthened the study. In the current study, we used same rest-interval durations for all groups. This showed that the fixed work/rest ratio may cause benefical effects on training adaptations [20]. Thus, further studies may investigate the effects of work/rest ratios to determine optimal work-rest intervals. Hematological parameters were not measured in the current study; however, due to inadequate hypoxic exposure duration in intermittent hypoxic training methods, it is unlikely that training with simulated hypoxic methods arouses useful hematological adaptations such as increased red blood cell numbers or total hemoglobin mass [40]. Lastly, detraining effects were not addressed in the current study and future research should investigate whether 15 s and 30 s SIT or SIH training for 4–6 weeks differ in the context of detraining.

## 5. Conclusions

In summary, we examined the effect of shortening the work-interval duration on training adaptations under hypoxic and normoxic conditions. Taken together, these results suggest that SIT is sufficiently able to increase anaerobic performance and reducing the work-interval duration of SIT does not diminish the increases in performance parameters. Also, it has been observed that a ~3500 m simulated normobaric hypoxic environment has no effect on SIT adaptations. Further research is needed to confirm these findings in higher altitudes such as 4000 m and in hypobaric hypoxic conditions.

## Figures and Tables

**Figure 1 ijerph-18-03976-f001:**
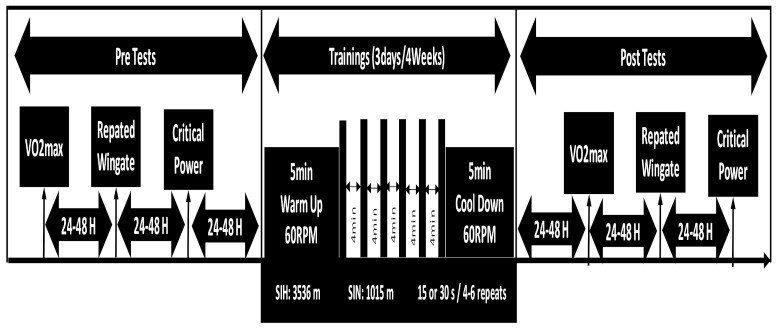
Schematic diagram of the study summary.

**Figure 2 ijerph-18-03976-f002:**
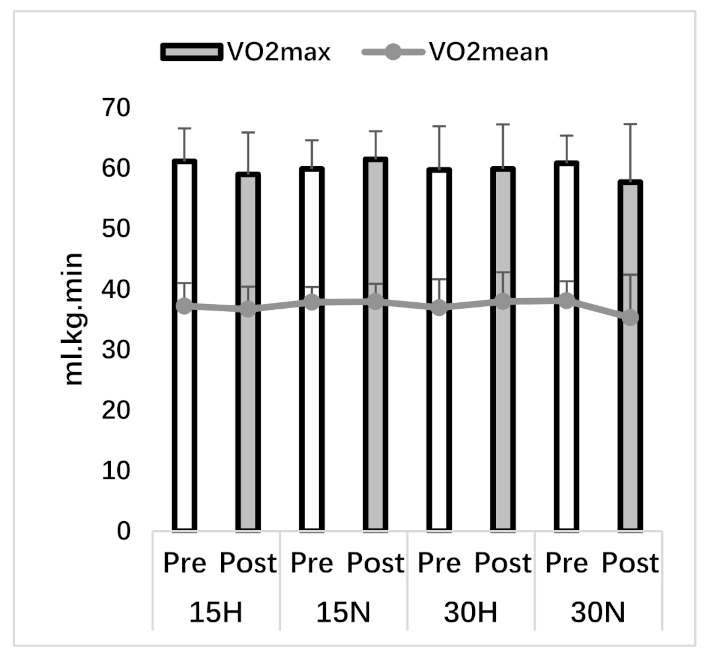
VO_2max_ and mean values.

**Figure 3 ijerph-18-03976-f003:**
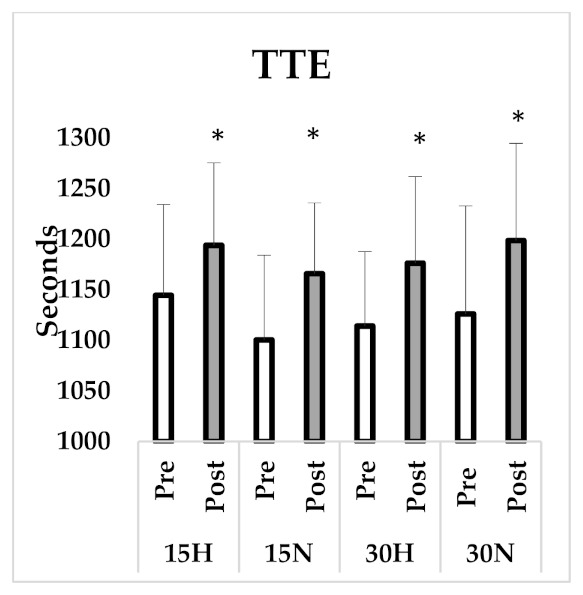
Time before exhaustion in VO_2max_ test. *: significant difference between pre and post test.

**Figure 4 ijerph-18-03976-f004:**
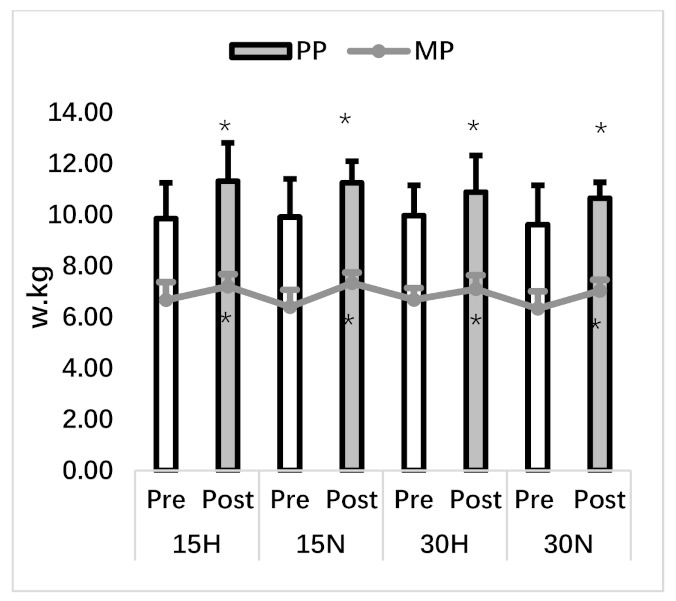
Repeated wingate test. PP: peak power, MP: mean power, *: significant difference between pre and post test.

**Figure 5 ijerph-18-03976-f005:**
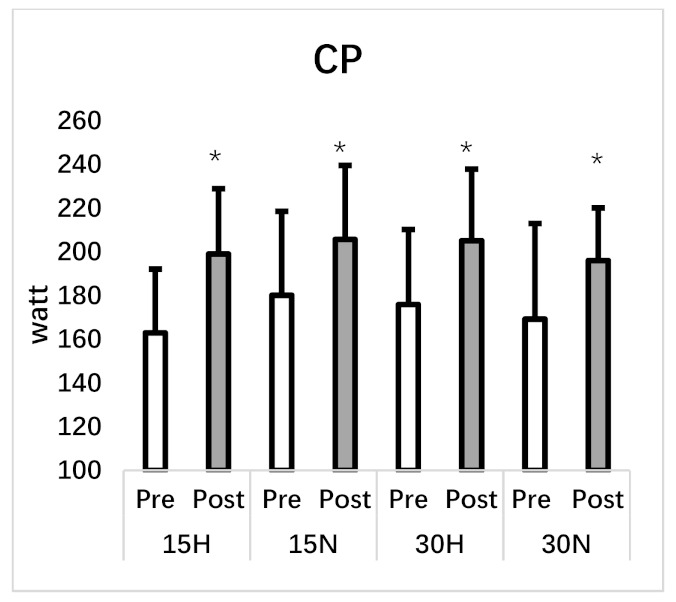
Critical power test. CP: critical power, *: significant difference between pre and post test.

**Table 1 ijerph-18-03976-t001:** Participant characteristics.

Parameters	15 H (*n* = 8)		15 N (*n* = 8)		30 H (*n* = 8)		30 N (*n* = 8)	
	Pre	Post	Pre	Post	Pre	Post	Pre	Post
Age (years)	20.75 ± 1.83		20.88 ± 1.46		20.50 ± 1.31		20.13 ± 1.25	
Height (cm)	177.63 ± 8.43		177.38 ± 4.75		176.88 ± 7.16		174.00 ± 7.54	
Body Weight (kg)	66.38 ± 7.75	66.90 ± 7.59	74.16 ± 9.11	75.11 ± 8.54	73.61 ± 6.75	75.11 ± 8.54	72.33 ± 11.10	72.49 ± 11.72
Body Fat Percent (%)	11.11 ± 3.80	11.50 ± 2.98	13.25 ± 3.54	14.10 ± 2.86	11.29 ± 4.09	11.83 ± 3.93	13.55 ± 4.28	13.55 ± 4.47
LBM (kg)	57.68 ± 6.52	59.13 ± 6.26	64.11 ± 5.91	64.38 ± 5.99	65.18 ± 4.76	64.95 ± 3.91	62.18 ± 7.26	58.85 ± 7.92
BMI	20.55 ± 1.52	21.14 ± 1.04	23.56 ± 2.70	23.85 ± 2.42	23.55 ± 2.09	23.66 ± 2.13	23.85 ± 2.92	22.69 ± 4.04

LBM: Lean body mass; BMI: Body mass index.

**Table 2 ijerph-18-03976-t002:** Maximal oxygen consumption test values with related parameters.

Parameters	15 H		15 N		30 H		30 N	
	Pre	Post	Pre	Post	Pre	Post	Pre	Post
VO_2mean_ (ml.kg.min)	37.23 ± 3.80	36.71 ± 3.73	37.88 ± 2.52	37.97 ± 2.94	36.96 ± 4.69	37.99 ± 4.82	38.13 ± 3.20	35.32 ± 7.09
VO_2max_ (ml.kg.min)	61.15 ± 5.45	59.00 ± 6.94	59.90 ± 4.73	61.49 ± 4.64	59.74 ± 7.23	59.91 ± 7.36	60.84 ± 4.56	57.72 ± 9.59
HR at VO_2max_	188.17 ± 8.67	188.10 ± 7.50	190.13 ± 6.14	195.55 ± 5.99	191.81 ± 5.78	191.59 ± 9.31	193.39 ± 10.22	194.08 ± 10.05
TTE (s)	1144.5 ± 89.86	1193.88 ± 81.54 *	1100.38 ± 83.80	1165.87 ± 69.99 *	1114.13 ± 73.76	1176.25 ± 85.61 *	1126.13 ± 106.8	1198.63 ± 96.09 *
RPEmean	11.92 ± 0.91	10.55 ± 1.68 *	11.60 ± 1.03	11.43 ± 1.30 *	11.58 ± 2.08	11.09 ± 1.66 *	11.91 ± 1.26	11.72 ± 1.11 *
RPEpeak	19.25 ± 0.88	18.50 ± 2.13	19.50 ± 0.75	19.12 ± 0.99	18.50 ± 2.82	18.87 ± 1.80	18.87 ± 1.55	19.62 ± 0.74
LA (mmol/L)	10.76 ± 4.21	13.51 ± 3.79 *	13.18 ± 3.62	15.69 ± 2.59 *	11.33 ± 2.74	15.48 ± 3.58 *	12.00 ± 2.97	13.84 ± 2.43 *

HR: heart rate; TTE: time to exhaustion; RPE: ratings of perceived exertion; LA: capillary blood lactate; *: significant difference between pre and post test *p* < 0.05.

**Table 3 ijerph-18-03976-t003:** Repeated wingate and critical power tests.

Parameters	15 H		15 N		30 H		30 N	
	Pre	Post	Pre	Post	Pre	Post	Pre	Post
PP (w/kg)	9.85 ± 1.40	11.32 ± 1.49 *	9.91 ± 1.49	11.25 ± 0.84 *	9.97 ± 1.18	10.89 ± 1.43 *	9.62 ± 1.53	10.65 ± 0.62 *
MP (w/kg)	6.67 ± 0.70	7.20 ± 0.48 *	6.40 ± 0.67	7.33 ± 0.41 *	6.68 ± 0.46	7.10 ± 0.54 *	6.33 ± 0.68	7.03 ± 0.43 *
Work (kJ)	12.22 ± 1.07	13.14 ± 1.93 *	13.41 ± 2.80	15.13 ± 2.41 *	13.29 ± 1.37	14.55 ± 1.49 *	12.80 ± 1.67	14.08 ± 2.08 *
Sprint Decrement (%)	65.47 ± 9.65	72.24 ± 9.07	67.52 ± 9.21	68.36 ± 5.97	68.38 ± 3.45	70.55 ± 5.73	64.06 ± 7.69	64.98 ± 6.62
sDec (%)	16.63 ± 7.24	11.52 ± 2.74 *	21.02 ± 5.97	13.49 ± 5.94 *	20.14 ± 6.39	14.30 ± 3.74 *	15.30 ± 6.91	11.85 ± 5.95 *
Average of Pre Sprint MPA	4.25 ± 0.85	2.05 ± 1.11 *	4.07 ± 1.37	3.01 ± 1.15 *	4.48 ± 2.33	3.14 ± 1.47 *	4.81 ± 1.21	2.89 ± 1.02 *
Average of Post Sprint MPA	6.91 ± 1.52	4.18 ± 1.40 *	6.80 ± 1.05	5.58 ± 1.91 *	5.52 ± 1.57	4.16 ± 2.00 *	6.78 ± 1.11	4.82 ± 1.29 *
Average of Pre Sprint RPE	12.50 ± 1.18	9.17 ± 2.20 *	12.65 ± 1.48	11.34 ± 1.61 *	12.04 ± 2.43	10.43 ± 2.39 *	13.97 ± 1.77	10.71 ± 1.89 *
Average of Post Sprint RPE	16.15 ± 2.30	13.00 ± 2.25 *	17.40 ± 1.64	15.46 ± 1.52 *	15.26 ± 2.80	14.00 ± 3.12 *	15.96 ± 1.06	14.31 ± 1.76 *
Average of Pre Sprint HR	144.90 ± 7.68	134.09 ± 11.38	145.75 ± 15.70	146.21 ± 10.99	142.30 ± 11.81	140.38 ± 7.86	143.28 ± 10.86	140.80 ± 14.95
Average of Post Sprint HR	185.09 ± 5.82	179.71 ± 6.09	184.22 ± 9.78	186.34 ± 7.25	183.31 ± 5.24	184.02 ± 5.67	184.18 ± 6.73	182.50 ± 10.40
Critical Power (w)	162.95 ± 29.15	198.98 ± 29.90 *	180.11 ± 38.41	205.67 ± 33.84 *	175.87 ± 34.34	205.09 ± 32.79 *	169.24 ± 43.71	196.04 ± 24.07 *

PP: peak power; MP: mean power; sDec: sprint decrement; MPA: muscle pain; RPE: ratings of perceived exertion; HR: heart rate; RPE: ratings of perceived exertion. *: significant difference between pre and post test *p* < 0.05.

**Table 4 ijerph-18-03976-t004:** Average of 4 weeks (12 sessions in total) of training parameters.

Parameters	15 H	15 N	30 H	30 N
PP (w/kg)	10.54 ± 0.95	10.87 ± 1.18	9.91 ± 1.27	9.97 ± 0.79
MP (w/kg)	8.21 ± 0.54 *	8.30 ± 0.62 *	6.66 ± 0.56	6.54 ± 0.43
Sprint Decrement (%)	47.08 ± 9.01	47.86 ± 9.89	68.96 ± 4.97 #	65.40 ± 4.09 #
sDec (%)	10.65 ± 2.56	10.76 ± 2.27	16.53 ± 2.85 #	14.79 ± 2.68 #
Work (kJ)	8.07 ± 1.21	9.14 ± 1.49	13.51 ± 1.51 #	13.09 ± 1.74 #
Average of Pre Sprint MPA	2.67 ± 0.67	2.84 ± 0.90	3.14 ± 0.77	2.82 ± 1.23
Average of Post Sprint MPA	4.43 ± 1.22	4.67 ± 1.34	5.01 ± 1.14	4.48 ± 1.31
Average of Pre Sprint RPE	9.93 ± 1.65	10.62 ± 1.22	11.12 ± 1.75	10.88 ± 2.08
Average of Post Sprint RPE	12.80 ± 2.35	13.81 ± 1.79	15.00 ± 1.72	14.09 ± 1.81
Average of Pre Sprint HR (beats/min)	143.48 ± 10.0	143.68 ± 11.7	142.39 ± 6.92	140.36 ± 12.1
Average of Post Sprint HR (beats/min)	173.91 ± 6.30	175.11 ± 8.90	180.23 ± 5.36	180.93 ± 9.41
Average of Pre Sprint SpO_2_ (%)	84.71 ± 3.11 $	92.26 ± 1.36	83.9 ± 2.51 $	91.67 ± 1.91
Average of Post Sprint SpO_2_ (%)	83.97 ± 2.58 $	90.61 ± 1.04	83.59 ± 1.72 $	90.16 ± 1.87

PP: peak power; MP: mean power; sDec: sprint decrement; MPA: muscle pain; RPE: ratings of perceived exertion; HR: heart rate; SpO_2_: arterial oxygen saturation; *****: significantly different from 30 H and 30 N; **#**: significantly different from 15 H and 15 N; $: significantly different from normoxia groups.

## Data Availability

The data presented in this study are available on request from the corresponding author. The data are not publicly available due to restrictions privacy.

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
