# Peer review of "The Effects of 15 or 30 s SIT in Normobaric Hypoxia on Aerobic, Anaerobic Performance and Critical Power"

_ijerph, 2021, doi:10.3390/ijerph18083976_

Round 1

Reviewer 1 Report

Congratulations on the work done and that has been contributed. An interesting design and a well-conducted experiment.

Just some consideration to keep in mind and/or change:

1. Table 1 uses the term Body Mass and not Body Weight. Which is the reason? The reason is because the experiment is carried out in altitude?

2. I recommend presenting figure number 1 using a clearer and simpler diagram (for example, there are words that it is difficult to read because the letter is too small).

3. A more precise explanation is needed of the way in which the 4 groups were configured, that is, how the distribution of subjects was carried out in each experimental group.

4. Was the lactate test performed only once at the end of the test, or several measurements over several minutes? (1, 3, 5 ...). A more precise explanation is needed.

5. Statistics section. It should be explained in detail how the comparative analysis was carried out, that is, what data were compared: only pretest and posttest of each of the 4 groups separately, or pre and post between the two groups of 15 and the two groups of 30 ?, or pre and post between groups H and N ?, or pre and post between the 4 groups? ... In general terms, provide a deep explanation to provide clarity to the reader about what analyzes and comparisons were made later.

6. Table 2 and Table 3 must be adjusted in the format of the cells (and homogeneous) to make it easier to understand the data.

7. Finally the duration of 4 weeks must be explained and justified (lines 91-92, line 119). It is very important to justify it because the entire article is based on that time period and, furthermore, it has not been considered as a limitation to the study.

Author Response

Congratulations on the work done and that has been contributed. An interesting design and a well-conducted experiment.

Just some consideration to keep in mind and/or change:

We would like to thank you for reviewing our article in detail, giving revisions and making our paper higher quality. We have responded to each comment in a point by point fashion.

  1. Table 1 uses the term Body Mass and not Body Weight. Which is the reason? The reason is because the experiment is carried out in altitude?

All body weight terms were replaced with body mass.

  1. I recommend presenting figure number 1 using a clearer and simpler diagram (for example, there are words that it is difficult to read because the letter is too small).

Figure 1 was resized.

  1. A more precise explanation is needed of the way in which the 4 groups were configured, that is, how the distribution of subjects was carried out in each experimental group.

4 groups were configured based on their pre training period VO2max values and this explanation was added to that sentence.

  1. Was the lactate test performed only once at the end of the test, or several measurements over several minutes? (1, 3, 5 ...). A more precise explanation is needed.

Lactate test was perfomed only immediately after test. (Line : 152)

  1. Statistics section. It should be explained in detail how the comparative analysis was carried out, that is, what data were compared: only pretest and posttest of each of the 4 groups separately, or pre and post between the two groups of 15 and the two groups of 30 ?, or pre and post between groups H and N ?, or pre and post between the 4 groups? ... In general terms, provide a deep explanation to provide clarity to the reader about what analyzes and comparisons were made later.

2 way analyses of variance was conducted to compare 1-) the pre to post training effect, 2-) group effect 3-) group x time effect and all these parameters were explained in the statistics and result section.

  1. Table 2 and Table 3 must be adjusted in the format of the cells (and homogeneous) to make it easier to understand the data.

Due to the much data exists, it can not be revised to otherwise tables do not fit to paper.

  1. Finally the duration of 4 weeks must be explained and justified (lines 91-92, line 119). It is very important to justify it because the entire article is based on that time period and, furthermore, it has not been considered as a limitation to the study.

Majority of studies in the literature performed with 4 weeks of training duration and mentioned in the introduction. Further, not to find any differences in VO2max and critical power in the current study was discussed in terms of training duration (4 weeks) in discussion section.

Reviewer 2 Report

Summary

This study aims to determine differences in SIT work intervals crossed with differences in normoxia/hypoxia as the authors identified this as a gap in the literature. The study has 8 participants per group perform either 15 or 30s SIT (both 4 – 6 reps) for 4 weeks, 3 days/week. They found anaerobic and critical power increased from SIT regardless of whether SIT was performed in normoxia or hypoxia, or whether the work interval was 15 or 30s.

Study Importance

Authors identified there was a gap in the literature on differences in work intervals when exercising in normoxia or hypoxia.

General Comments

The manuscript could benefit from a proficient English user reviewing sentence structure as some word choices are odd and make some sentences difficult to read. Examples include the use of “Seemingly” and “Besides.” There also seem to be words missing in a sentence or improper use of (or lack thereof) punctuation or verb tenses.

Since normobaric hypoxia did not yield a further increase in performance, what does this mean for the manuscript?

Were there any statistical differences between groups for pre-test results?

Why was the VO2max test conducted on a treadmill when all other test sessions and training sessions were conducted on a cycle ergometer? Since the VO2max test was not conducted on a cycle ergometer, do you think you might have gotten different results if you had? Considering the specificity principle, shouldn't this be included in the discussion as a possible limitation? If the purpose of the test was to achieve a VO2peak, is it a VO2peak test and not a VO2max test?

Tables had odd variable placements in the upper left-hand corner. Recommend revising. Recommend replacing the , in the number with a . for ease of reading. Ensure tables are all on the same page for ease of reading.

Some abbreviations are not previous defined before their use in either the methods section or results section such as VAS and TTE.

Specific Comments

Line 19: missing a word or two. “… new concept that has been shown to enhance…” If SIT was first used two decades ago, is it really new?

Line 21: saying SIT training is like saying sprint interval training training since the “T” in SIT stands for training. Recommend changing throughout and in the title

Line 24-25: missing a word. “… was found to be significantly longer in the…”

Line 26: need the p-value for the wingate test significantly increasing from pre- to post-training like the other results stated

Line 71-75: seems to be more of a discussion point rather than an introductory point as the reader has not read the methods yet. It would be best to point out the gaps in the current literature in the intro not how the current study fills the gaps

Line 85-91: seems the sentences from 87-91 just repeat what was said in 85-86

Table 1. the variable label body composition seems to be out of place

Line 106: what were the reasons for participant withdrawal from the study? Lack of time, injury from study/outside of study, unable to contact participants, etc.?

2.1 Participants paragraph: this paragraph has many grammar/punctuation errors. Recommend revising.

Line 125: Figure titles go below the figure. Ensure consistent throughout

Line 129: How were the participants blinded to their exercise performed or environment? Wouldn’t the participants be aware they were performing 15 or 30s work intervals or were in normoxia or hypoxia?

Line 138: were the training sessions carried out on a treadmill or bike?

Line 141: was the session duration different between hypoxia conditions and normoxia differences? Is that what this was specified in this line for hypoxia? What were the durations of the normoxia training sessions?

Line 218: Why was Fisher’s LSD used for the post-hoc test rather than Tukey’s or Bonferroni? Statisticians argue against it’s use because it does not correct for the multiple comparisons made (in this studies case comparison of 4 groups (15H, 15N, 30H, 30N) if I am interpreting the statistics section correctly. Tukey’s or Bonferroni would be a better choice.

Line 222: Results section would benefit from subheadings (VO2max test outcomes, repeated wingate test outcomes, critical power test outcomes).

Line 382-386: Though I agree identifying the lack of hypobaric hypoxia and not monitoring nutrition were limitations of the study, the statements are vague and will leave the reader wondering why these are limitations of the study. More detail needs to be provided for why these are limitations. Should the fact that normobaric hypoxia has been argue to not fully simulate natural hypoxia or not be similar to hypobaric hypoxia be discussed as a possible influencer of the results? For nutrition, any participants taking any supplements that may improve anaerobic performance such as creatine as recorded in the 24-hr food logs?

Author Response

This study aims to determine differences in SIT work intervals crossed with differences in normoxia/hypoxia as the authors identified this as a gap in the literature. The study has 8 participants per group perform either 15 or 30s SIT (both 4 – 6 reps) for 4 weeks, 3 days/week. They found anaerobic and critical power increased from SIT regardless of whether SIT was performed in normoxia or hypoxia, or whether the work interval was 15 or 30s.

Study Importance

Authors identified there was a gap in the literature on differences in work intervals when exercising in normoxia or hypoxia.

General Comments

The manuscript could benefit from a proficient English user reviewing sentence structure as some word choices are odd and make some sentences difficult to read. Examples include the use of “Seemingly” and “Besides.” There also seem to be words missing in a sentence or improper use of (or lack thereof) punctuation or verb tenses.

Since normobaric hypoxia did not yield a further increase in performance, what does this mean for the manuscript?

First of all, we would like to express our respects and thank you for your contribution (with your deep knowledge and experience in this field) to this paper which is the first in this topic. Our revisions are as follows.

Not yielding a further increase by normobaric hypoxia in performance was discussed in the discussion section as follows:

1-) The fact that our participants have a training background and a high baseline VO2peak level (~60ml.kg-1.min-1) may have caused this situation. Thus, 12 sessions of SIH and SIN have may not provided enough additional training stimuli to increase VO2peak in this population. (Line: 313-316)

2-) Indeed, in a study [6] conducted at different altitudes (between FiO2=13.0–20.9=) were found to provide higher PP and lactate-related adaptation gains than the same training routine at sea level while training at lower altitudes not providing additional gains. (Line: 324-327)

Were there any statistical differences between groups for pre-test results?

No, due to the 4 groups divided in a counter-balanced manner, no significant differences was observed in the beginning of the study.

Why was the VO2max test conducted on a treadmill when all other test sessions and training sessions were conducted on a cycle ergometer? Since the VO2max test was not conducted on a cycle ergometer, do you think you might have gotten different results if you had? Considering the specificity principle, shouldn't this be included in the discussion as a possible limitation? If the purpose of the test was to achieve a VO2peak, is it a VO2peak test and not a VO2max test?

Because participants of this study comprised of team field players (football,handball, basketball), we prefered to treadmill VO2max test to simulate running based activities of participants. On the other hand, the training sessions were carried out on an cycle ergometer, so there was a strong adaptation to this type of work by the locomotor system. In this way, the possible better efficiency of work on a cycle ergometer was eliminated, resulting not from physiological and biochemical changes, but from the mechanical work of muscles. From this reason, it would be expected that likely significant improvement in VO2max values can be observed in the treadmill test. Both test and values are VO2max datas, values seen as VO2peak was written mistakenly and revised.

Tables had odd variable placements in the upper left-hand corner. Recommend revising. Recommend replacing the , in the number with a . for ease of reading. Ensure tables are all on the same page for ease of reading.

These directions were performed.

Some abbreviations are not previous defined before their use in either the methods section or results section such as VAS and TTE.

Yes, VAS and TTE were mistakenly forgotten. Abbreviations were defined before used.

Specific Comments

Line 19: missing a word or two. “… new concept that has been shown to enhance…” If SIT was first used two decades ago, is it really new?

Sentence was revised and “new” phrase was erased.

Line 21: saying SIT training is like saying sprint interval training training since the “T” in SIT stands for training. Recommend changing throughout and in the title

“Training” phrase was erased throughout the paper.

Line 24-25: missing a word. “… was found to be significantly longer in the…”

Sentence was revised.

Line 26: need the p-value for the wingate test significantly increasing from pre- to post-training like the other results stated

P=0.01 was added.

Line 71-75: seems to be more of a discussion point rather than an introductory point as the reader has not read the methods yet. It would be best to point out the gaps in the current literature in the intro not how the current study fills the gaps

Sentence beginning with “our study … “ was removed and rest of paragraph was revised.

Line 85-91: seems the sentences from 87-91 just repeat what was said in 85-86

Line 85-86 was removed.

Table 1. the variable label body composition seems to be out of place

Body composition phrase was removed.

Line 106: what were the reasons for participant withdrawal from the study? Lack of time, injury from study/outside of study, unable to contact participants, etc.?

3 of them declared that lack of time to continue the study and 5 participants left without any reason. An explanation was added to line 106-107.

2.1 Participants paragraph: this paragraph has many grammar/punctuation errors. Recommend revising.

That section was revised.

Line 125: Figure titles go below the figure. Ensure consistent throughout

Titles of figures were moved to below and revised throughout.

Line 129: How were the participants blinded to their exercise performed or environment? Wouldn’t the participants be aware they were performing 15 or 30s work intervals or were in normoxia or hypoxia?

Participants only get blinded to environment supplied via hypoxia generator of which altitude level can be changed manually and can only be seen by researcher. Further, 15 and 30s work interval groups were seperated to else laboratory but participants were aware of the work-interval durations.

Line 138: were the training sessions carried out on a treadmill or bike?

Sentence was revised.

Line 141: was the session duration different between hypoxia conditions and normoxia differences? Is that what this was specified in this line for hypoxia? What were the durations of the normoxia training sessions?

Sentence was revised.

Line 218: Why was Fisher’s LSD used for the post-hoc test rather than Tukey’s or Bonferroni? Statisticians argue against it’s use because it does not correct for the multiple comparisons made (in this studies case comparison of 4 groups (15H, 15N, 30H, 30N) if I am interpreting the statistics section correctly. Tukey’s or Bonferroni would be a better choice.

Due to the some studies used Fisher’s LSD, We preferred this post hoc analyse, However, we revised the post hoc analyse and replaced with Bonferroni.

Line 222: Results section would benefit from subheadings (VO2max test outcomes, repeated wingate test outcomes, critical power test outcomes).

Results sections was divided into subheadings.

Line 382-386: Though I agree identifying the lack of hypobaric hypoxia and not monitoring nutrition were limitations of the study, the statements are vague and will leave the reader wondering why these are limitations of the study. More detail needs to be provided for why these are limitations. Should the fact that normobaric hypoxia has been argue to not fully simulate natural hypoxia or not be similar to hypobaric hypoxia be discussed as a possible influencer of the results? For nutrition, any participants taking any supplements that may improve anaerobic performance such as creatine as recorded in the 24-hr food logs?

All limitation section was revised as you directed.

Round 2

Reviewer 2 Report

Thank you for taking the time to review and make changes based on the previous comments. It is clear the authors have put in the time to improve the manuscript.

Figure 1. ensure it is all on one page, title included